# Targeting PI3K/AKT/mTOR Pathway in Breast Cancer: From Biology to Clinical Challenges

**DOI:** 10.3390/biomedicines11010109

**Published:** 2023-01-01

**Authors:** Krisida Cerma, Federico Piacentini, Luca Moscetti, Monica Barbolini, Fabio Canino, Antonio Tornincasa, Federica Caggia, Sara Cerri, Alessia Molinaro, Massimo Dominici, Claudia Omarini

**Affiliations:** 1Department of Medical and Surgical Sciences for Children & Adults, University Hospital of Modena, 41122 Modena, Italy; 2Division of Medical Oncology, University Hospital of Modena, 41122 Modena, Italy; 3GOIRC (Gruppo Oncologico Italiano di Ricerca Clinica), 43126 Parma, Italy

**Keywords:** PI3K, breast cancer, PI3K inhibitors, mTOR inhibitor, everolimus, alpelisib, AKT

## Abstract

Breast cancer (BC) is the most common women cancer and cause of cancer death. Despite decades of scientific progress in BC treatments, the clinical benefit of new drugs is modest in several cases. The phosphatidylinositol 3-kinase (PI3K)/protein kinase B (AKT)/mammalian target of rapamycin (mTOR) pathway mutations are frequent in BC (20–40%) and are significant causes of aggressive tumor behavior, as well as treatment resistance. Improving knowledge of the PI3K/AKT/mTOR pathway is an urgent need. This review aims to highlight the central role of PI3K-mTORC1/C2 mutations in the different BC subtypes, in terms of clinical outcomes and treatment efficacy. The broad base of knowledge in tumor biology is a key point for personalized BC therapy in the precision medicine era.

## 1. Introduction

Breast cancer (BC) remains the most common cancer diagnosed in women. Despite the increase of knowledge in cancer biology and treatment, BC is the fifth cause of cancer mortality worldwide [1]. 

The development of precision medicine for the management of BC is an appealing concept; however, major scientific and logistical challenges hinder its implementation in the clinic. The identification of mutational drivers remains the biggest challenge, because, with few exceptions, such as estrogen receptor (ER) or human epidermal growth factor receptor 2 (HER2), no other validated oncogenic drivers of BC tumorigenesis exist.

Of note, the phosphatidylinositol 3-Kinase (PI3K)/protein kinase B (AKT)/mammalian target of rapamycin (mTOR) pathway is an important signaling pathway involved in treatment failure. The main causes of pathway deregulation are PIK3CA mutations, AKT mutations or tensin homolog (PTEN) loss [2]. Pathway dysregulations are known mechanisms of endocrine resistance, as well as anti-HER2-targeted agent resistance [3,4,5,6,7]. PI3K/AKT/mTOR mutations are more frequent in hormone receptor-positive (HR+) BC compared to other BC subtypes [5].

For that reason, several targeted drugs are under investigation in order to restore PI3K/AKT/mTOR pathway activity. Currently, there are only two targeted agents approved for the treatment of MBC patients, both in HR+/HER2-negative (HER2−) disease. The first one is Everolimus, an mTOR inhibitor, approved in combination with Exemestane, based on the BOLERO 2 trial results [8,9]. The second one is Alpelisib, recently approved in combination with Fulvestrant in endocrine-resistant PI3K-mutated HR+ metastatic (M)BC [10]. 

This review aims to point out the role of PI3K/AKT/mTOR pathway mutations in the different BC subtypes, with a focus on their clinical impact in terms of survival outcomes and treatment efficacy.

## 2. PI3K/AKT/mTOR Pathway

The PI3K/AKT/mTOR pathway is physiologically involved in cell metabolism, growth, proliferation and apoptosis [11] through the activation of tyrosine kinases receptor (RTK) and G-protein-coupled receptors [12].

In particular, PI3Ks are a family of intracellular kinases subdivided into three classes (class I, II and III) according to their sequence homologies and in vitro substrate preference [13]. Class I is the major PI3K family of enzymes. It is further subdivided into class IA and class IB based on the activated receptors [14]. Class IA is composed of different catalytic and regulatory sub-units that directly interact with several tyrosine kinases receptors, such as the epidermal growth factor receptor (EGFR), platelet-derived growth factor receptor (PDGFR) and insulin-like growth factor-1 receptor (IGF-1R) [15]. The IA-PI3Ks are activated by cell surface receptors, such as G protein-coupled receptors, RTKs and the small G protein RAS [14]. On the contrary, small molecules, such as GTPases, activated the class II enzymes [15]. The central mediator of the PI3K pathway is the AKT that activates over 100 substrates, including mTOR [16]. By contrast, PTEN is the main negative regulator of PI3K signaling [17] (Figure 1). During cancer development, alterations in the PI3K/AKT/mTOR pathway are mainly due to PIK3CA and AKT mutations, RTKs overexpression or PTEN loss [18]. Of note, PTEN and AKT1mutations are mutually exclusive mutations [19,20].

As mentioned before, PI3K/AKT/mTOR mutations are prevalent in HR+ tumors compared to other BC subtypes (Figure 2). These mutations accord approximately in 40% of HR+ BC, mainly in the helical and kinase domains of the PIK3CA gene, including three main hotspot mutations: exon 9 E545K or E542K and exon 20 H1047R [21]. These gene alterations activate PI3K enzyme, leading to constitutive phosphorylation of AKT and its downstream effecttors [22]. Mutations in other components of the pathway are less common and include mutations in AKT1 (2–3%) and PI3K regulatory subunit α (1–2%), loss-of-function mutations in PTEN (2–4%) and mutations or overexpression of RTKs [20,23].

In HER2-positive (HER2+) disease, PIK3CA mutations occur in nearly 25% of the BC and represent a well-known mechanism of acquired resistance to HER2-targeted therapies [24]. By contrast, in the TNBC subgroup, the most frequent mutation is PTEN loss (about 30–50%), while PIK3CA mutations occur in less than 10% of cases, principally in androgen receptor-positive TNBC [25]. 

### 2.1. Tests

PIK3CA mutations can arise early in tumorigenesis, as well as can be acquired during disease progression [26]. Available tests for evaluating PI3K status are the Polymerase Chain Reaction (PCR) single-gene test or the Next-Generation-Sequencing (NGS), both validated in different platforms [27]. Comparative effectiveness research indicated that NGS analysis seems to be more sensitive than PCR-based assays [28]. In a retrospective analysis conducted in the SOLAR-1 population, the *PIK3CA* mutations detected by PCR were 60% compared to 71% identified with NGS. In fact, the NGS technology is more sensitive than PCR in the detection of less common spectrums of PI3K alterations [29]. PIK3CA mutations can be detected in both tissue and/or plasma specimens. Quite high concordance using formalin-fixed paraffin-embedded (FFPE) tissue- based and plasma testing has been observed. Additional analysis from BELLE-2 and BELLE-3 trials using the BEAMing PCR assay showed a similar concordance of PIK3CA mutation status between circulating tumor (ct)DNA and tumor tissue analysis (77% and 83%, respectively) [30,31]. Similar findings came from another retrospective study conducted by Chae et al., where the concordance between ctDNA and tumor tissue using an NGS-based assay was about 75% [32]. The different PI3K expressions between plasma and tissue specimens reported (about 25% of cases) was justified by both high tumor heterogeneity and sample contaminations [33]. In fact, recent evidence seems to suggest that the PI3K alterations assessed by liquid biopsy better reflect tumor biology and patient prognosis. In the SOLAR-1 population, patients with PIK3CA mutations in tissue samples had a 35% of risk reduction in disease progression compared to 45% for those with PIK3CA mutations detected in ctDNA [32]. The available evidence and the easy accessibility of ctDNA compared to tumor biopsy makes the ctDNA the possible future primary approach [34]. 

### 2.2. PI3K/AKT/mTOR Targeted Agents

The class of PI3K/AKT/mTOR targeted agents includes different drugs classified according to their mechanisms of action (Figure 1). Drugs mainly investigated in BC are briefly listed below. Table 1 summarizes all the agents studied for BC treatment.
pan-PI3K inhibitors (PI3Kis)

The Pan-PI3K inhibitors inhibited the kinase activity of the four isoforms of class I PI3Ks: α, β, γ and δ. Preclinical models have reported that suppression of PI3K-signaling pathway restored endocrine sensitivity [35]. The activity of PI3Kis was firstly evaluated as single agent in MBC patients showing few treatment benefits with high frequency of class-specific adverse events (AEs) [36]. 

**Buparlisib** is an oral 2,6-dimorpholino pyrimidine derivative that acts as a potent pan-PI3K inhibitor. It showed efficacy against p110α somatic mutations frequently detected in human cancers, but it was minimally effective against the PI3K class III and class IV family members [37]. It was mainly investigated in BELLE2 and BELLE3 trials conducted in HR+ MBC. Data from the BELLE3 study suggested a moderate clinical benefit with Buparlisib plus endocrine therapy, but an important safety profile that limited the drug development. In particular, the most common toxicities were grade 3/4 hypertransaminasmia (40%), hyperglycaemia (12%), hypertension (6%) and fatigue (4%) [38]. Drug-related AEs were mood alterations with evidence of attempted suicides (2%), depression (1%) and anxiety (1%). 

**Pictilisib** (GDC-0941; Genentech, San Francisco, CA, USA) is an orally available small molecule that showed clinically significant activity in preclinical BC models [39]. Pictilisib combined with paclitaxel or Fulvestrant was evaluated in both PEGGY (NCT01740336) and FERGI (NCT01437566) trials. Patients in treatment with Pictilisib mainly experienced rash (4%), pneumonitis (3%), diarrhea (3%), deranged transaminases (3%), abdominal pain (2%) and stomatitis (2%) [40].
ii.Isoform-Specific PI3K Inhibitors


The Isoform-specific inhibitors link a specific PI3K isoform in order to reduce the toxicity of pan-PI3K inhibitors. In particular, the PI3Kα inhibitors selectively inhibit the class I PI3K catalytic subunit α isoform. Alpelisib and Taselisib showed positive results in clinical trials conducted in PIK3CA-mutated patients. 

**Alpelisib** (BYL719; Novartis Pharmaceuticals, Basel, Switzerland) is the first oral PI3Kα inhibitor acting against the subunit α isoforms [41]. It has shown synergistic antitumor activity when associated to endocrine therapy in HR+PIK3CA-mutated BC cells in preclinical and clinical phase III trial (SOLAR-1) [10].

The most common AEs are gastrointestinal disorders (73%), hyperglycemia (62%), fatigue (54%) and rash (42%). Most of the side effects are dose dependent and cumulative, and tend to appear in the first weeks of treatment [10].

**Taselisib** (GDC-0032, Genentech, San Francisco, CA) is an orally PI3Kα inhibitor targeting α, δ and γ isoforms [42]. The triplet of Taselisib plus Palbociclib and Fulvestrant was investigated in heavily pretreated PIK3CA-mutant HR+ MBC patients [43]. The most reported grade 3–4 adverse events are colitis (13.3%), diarrhea (11.7%), hyperglycemia (6.7%) and pneumonia (5%) [44].
iii.AKT Inhibitors


AKT is a downstream target of PI3K [45]. AKT has three isoforms, AKT 1, 2 and 3, which have similar structures and are directly linked by the AKT inhibitors. 

**MK-2206** is an orally bioavailable allosteric inhibitor of AKT (protein kinase B), binding the domain in a non-ATP competitive manner. It was investigated in early and advanced settings with modest results [46,47]. For those reasons, it was not further developed.

**Capivasertib** is an oral agent that binds and inhibits all AKT isoforms. It has demonstrated promising antitumor activity in a phase I study in patients with solid tumors harboring AKT1 E17K mutations [48]. The most common grade 3-4 AEs are diarrhea (22%), hyperglycemia (13%), neutropenia (11%) and maculopapular rash (9%) [49].

Ipatasertib is an orally bioavailable inhibitor of AKT [50]. It is currently under study in clinical trials conducted in HR+ MBC in combination with endocrine treatment and/or CD4/6 inhibitors, paclitaxel and immunotherapy (Table 2). The most reported AEs are diarrhea (23%), neutropenia (18%), peripheral neuropathy (7%), fatigue or asthenia (5%) and pneumonia (5%) [51].
iv.mTOR Inhibitors


mTOR is a serine/threonine specific kinase able to regulate cell proliferation and survival [52]. mTOR forms two multiprotein complexes, mTORC1 and mTORC2, that are targetable from cancer drugs [53]. 

**Everolimus** is an allosteric inhibitor of mTORC1. Based on the results of phase III BOLERO-2 study, the combination of Everolimus and Exemestane was approved for the treatment of HR+/ HER2− MBC progressed on endocrine treatment. Class-related AEs are stomatitis (8%), anemia (7%), pneumonitis (5%), hyperglycemia (5%) and fatigue (3%) [54].

**Temsirolimus** is a selective mTORC1 inhibitor [55]. It was evaluated in a phase II study at the dose of 25 mg weekly in heavily treated HR+ and/or HER2+ BC showing minimal activity [56]. On the contrary, in the phase III HORIZON trial, Temsirolimus, in addition to Letrozole, showed a significant advantage compared to ET alone in HR+ MBC patients. A high rate of grade 3 and 4 class-related AEs have been reported [57].

Knowledge on the PI3K/AKT/mTOR pathway is mandatory not only for developing targeted agents, but also for innovative treatment strategy. In particular, the role of mTOR as a metabolic or immune checkpoint regulator may open the opportunity for new therapeutic approaches [58,59]. For example, some AMP-activated protein kinase (AMPK) activators, such as metformin, have been studied and are under investigation in preclinical research due to their ability to stimulate PI3K/Akt and inhibit mTOR/S6K [60]. Available data on the anticancer activity of metformin in BC patients are still controversial, but promising [61]. Moreover, recent evidence showed that even PLD1, through Rheb, is involved in the activation of mTOR, suggesting an interesting mechanism of PLD-mTOR signaling cross talk [62].

### 2.3. Early Breast Cancer Neoadjuvant Setting

Even if the PI3K/AKT/mTOR-targeted agents have been investigated in a neoadjuvant setting, none has been approved due to the loss in gain in pathological complete response (pCR) rate and survival outcomes (Table 3). 

### 2.4. HR Positive EBC

Two phase II neoadjuvant trials, conducted in HR+/HER2− early BC patients, evaluated the efficacy of Taselisib and Alpelisib in combination with endocrine. In both trials, the pCR rate was the primary endpoint. In particular, the LORELEI trial, a randomized, double-blind, placebo-controlled study, investigated the combination of Taselisib plus Letrozole compared to Letrozole alone [63]. No statistically significant difference in pCR rate has been reported between the two groups, neither in the overall population, nor in the PI3K-mutated patients. The addition of Taselisib to Letrozole was associated with a higher proportion of objective response rate (ORR), independently from the PI3K status (39% in the placebo group vs. 50% in the Taselisib one, *p* = 0.049) [64]. Negative results in term of pCR and ORR were reported in the NEO-ORB study, where Letrozole was combined with Alpelisib as the primary treatment strategy [65]. 

### 2.5. Triple Negative Early BC

The phase II FAIRLANE trial explored the efficacy of Ipatasertib in addition to Paclitaxel vs. Paclitaxel alone in early TNBC [66]. The primary endpoint was the pCR rate in the overall population, PTEN-low population and PIK3CA/AKT1/PTEN-mutated tumors. In all subgroup analyses, the addition of Ipatasertib to chemotherapy showed only a trend in pCR rate in favor of sperimental arm: 17% vs. 13% in the overall population, 16% vs. 13% in the PTEN-low population and 18% vs. 12% in PIK3CA/AKT1/PTEN-altered tumors, respectively [66]. Following the pre-clinical evidence that PI3K pathway inhibitions lead to suppression of BRCA gene transcription through MEK1 and ERK activation, the combination of PI3K and PARP inhibitors have been studied [67,68,69,70]. 

### 2.6. HER2 Positive EBC

Considering that PIK3CA mutations lead to HER2-targeted agents resistance, the association between anti-HER2 therapy and PI3K/AKT/mTOR inhibitors has been widely tested [69]. A combination of MK-2206 with Paclitaxel and Trastuzumab was investigated in a HER2+ population enrolled in an I-SPY 2 trial. The reported pCR rate was 48% in the MK-2206 arm compared with 29% in the control one [48]. In contrast, no clinical advantage due to Buparlisib addiction to Paclitaxel and Trastuzumab was reported in the phase II NeoPHOEBE trial [71]. 

## 3. Metastatic Breast Cancer 

The activity of PI3K/AKT/mTOR-targeted agents seems to be more promising in a metastatic setting (Table 4). Actually, two PI3K/AKT/mTOR pathway inhibitors are approved for the treatment of metastatic (M) BC: Everolimus, in combination with Exemestane, in HR+/HER2− endocrine resistant MBC and Alpelisib, in combination with Fulvestrant in the case of PI3K-mutated tumors. 

### 3.1. HR Positive MBC

The first targeted agent approved for the treatment of HR+/HER2− MBC was Everolimus. The phase III BOLERO 2 trial showed PFS benefits due to the addition of Everolimus to Exemestane in endocrine resistance MBC (median progression free survival (mPFS) 10.6 months versus 4.1 months; hazard ratio (HR) 0.43; 95% CI: 0.35–0.54; *p* < 0.001). No gain in overall survival (OS) has been observed [57]. Moreover, Everolimus confirmed its activity combined with Tamoxifen too. In the TAMRAD trial, the combination strategy increased the clinical benefit rate from 42% to 61%, with an advantage of 4 months in time to progression [72]. Temsirolimus was the other mTOR inhibitor tested in a metastatic setting. No significant survival benefit has been reported in the overall population of the HORIZON trial [57]. In the subgroup analysis, patients younger than 65 years had a slight, but statistically significant PFS benefit (mPFS FS 9.0 months in Temsirolimus arm versus 5.6 months in endocrine therapy (ET) alone; HR 0.75; 95% CI: 0.60–0.93; *p* = 0.009) [57]. 

Based on the SOLAR 1 trial results, the FDA approved the use of Alpelisib with Fulvestrant for the treatment of HR+/HER2− PI3K-mutated MBC progressed on or after an endocrine-based regimen. In 2020, EMA approved the use of Alpelisib in association with Fulvestrant in HR+/HER2− PI3K-mutated MBC patients progressed on endocrine monotherapy [10]. The mPFS in the PIK3CA-mutant cohort was significantly improved with Alpelisib compared to endocrine treatment alone (mPFS 11.0 vs. 5.7 months, HR = 0.65; *p* = 0.00065). PIK3CA status was determined on both tumor tissue samples and plasma ctDNA. No benefit from Alpelisib addition was observed in the PIK3CA–non-mutant population. Of note, only 20 out of 572 patients enrolled were previously treated with a cyclin-dependent kinases 4 and 6 (CDK4/6) inhibitor [10]. The phase II trial (BYlieve), a multicenter open-label, non-comparative study, showed the efficacy of Alpelisib plus Fulvestrant in patients with PIK3CA-mutated HR +/HER2− BC also pre-treated with CDK4/6 inhibitors [73]. In the phase III SANDPIPER trial, the combination of Taselisib with Fulvestrant in endocrine-resistant PIK3CA-mutant patients significantly improved mPFS from 5.4 months to 7.4 months (stratified HR 0.70; 95% CI, 0.56–0.89; *p* = 0.0037) [74]. 

The BELLE trials investigated the benefit from the addition of Buparlisib to endocrine therapy or chemotherapy. In the phase III BELLE-2 trial, the safety and the efficacy of Buparlisib in combination with Fulvestrant was explored in postmenopausal women with aromatase inhibitor-resistant HR+/HER2− MBC, who had received at least one previous line of therapy for advanced disease [30]. A significant improvement in mPFS was observed in the Buparlisib arm versus placebo one (6.9 vs. 5.0 months; HR = 0.78; *p* = 0.00021). Patients were stratified according to PI3K status (activated vs. non-activated vs. unknown): women with a known PIK3CA mutation or an activated PI3K pathway had better mPFS compared to patients with unknown status [31]. The BELLE-3 trial investigated the power of Buparlisib to restore endocrine sensitivity in patients progressed after aromatase inhibitors (AI) or an mTOR inhibitor. Overall, the addition of Buparlisib to Fulvestrant improved the mPFS (3.9 vs. 1.8 mo; HR = 0.67; *p* = 0.00030), mainly in those with PIK3CA mutation detected by ctDNA analysis (mPFS 4.2 months, *p* = 0.00031) [30]. Data from both BELLE trials show that PIK3CA-mutant BCs had more benefit from the addition of PI3K inhibitors to endocrine therapy compared to PI3K wild type ones. On the contrary, the addition of Buparlisib to chemotherapy did not report any survival benefit in the phase II BELLE-4 population, regardless of the PIK3CA mutation status [75]. 

Pictilisib activity combined with Fulvestrant or Paclitaxel was investigated in two phase II trials (FERGI and PEGGY) conducted in HR+/HER2− endocrine resistant BC patients [42]. In both studies, no significant benefit in term of PFS has been found [42]. Promising results have been reported for Capivasertib added to Fulvestrant in patients progressed after AI (phase II FAKTION trial). The Capivasertib plus Fulvestrant strategy showed more than 5 months in PFS benefit compared to Fulvestrant alone (10.3 months versus 4.8 months; *p* = 0.0018, respectively). PI3K mutation did not affect the sensitivity to Capivasertib [76]. Results from the phase III trial CAPItello-291 are still in progress [77]. Less effective seems to be the combination of Capivasertib and Paclitaxel (BEECH trial) [49]. 

### 3.2. Triple Negative MBC

The PI3K/AKT/mTOR inhibitors activity is still at early-phase development in metastatic TNBC. In particular, data from two phase II trials showed promising results for the AKT inhibitors class. In LOTUS trial, Ipatasertib was tested with weekly Paclitaxel, showing a statistically significant PFS advantage independently of PTEN status (6.2 versus 4.9 months, respectively; *p* = 0.037) [51]. Moreover, in another phase II trial, patients with mutation in the PIK3CA/AKT1/PTEN pathway had a significant PFS improvement with Ipatasertib compared to placebo (9.0 months vs. 4.9 months, HR 0.44, 95% CI 0.20–0.99, *p* = 0.041). The phase III trial (IPATunity130) conducted in HER2− MBC with a known PIK3CA/AKT1/PTEN-altered pathway is actually ongoing [78]. Capivasertib in combination with Paclitaxel was evaluated in a PAKT trial, showing a possible benefit in patients with genetic alterations of PIK3CA, AKT1 or PTEN. In particular, in the Capivasertib cohort ORR, clinical benefit rate, mPFS and mOS were 35.3%, 52.9%, 9.3 months and not reached, respectively, compared to 18.2%, 27.3%, 3.7 months and 10.4 months in the placebo arm [79]. 

Based on the preclinical data that reported less activity of PD-L1 blockade agents in PTEN loss cells, trials investigating the combination of PIK3β inhibitors and anti-PD-L1 therapies have been setup (Table 2) [80]. Similar negative results for the association of PI3k/AKT/mTOR inhibitors and chemotherapy have been reported in triple negative BC patients. No improvement survival outcomes for Buparlisib plus Paclitaxel (BELLE-4 trial) and mTOR inhibitors plus liposomal doxorubicin and Bevacizuamb [81]. 

Finally, data from phase I-II suggest the potential benefit of PI3k inhibitors combined with different targeted agents (i.e., Palbociclib, Enzalutamide and Olaparib) [82,83,84]. In particular, the combination of a PI3K inhibitor and a PARP inhibitor seems to reduce BRCA1/2 expression, increasing the antitumor effects of Olaparib [66,67,68,69,70,84]. 

### 3.3. HER2 Positive MBC

Preclinical evidence supported the involvement of the PI3K/AKT/mTOR pathway in the mechanism of HER2 resistance [85,86]. These finding are the rationale for combining PI3K/AKT/mTOR inhibitor agents with anti HER2-targeted therapies [87]. Both BOLERO-1 and BOLERO-3 trials evaluated the efficacy of Everolimus and Trastuzumab, showing only a modest PFS advantage (mPFS 7 months vs. 5.78 months) [88,89]. Disappointing results have also been reported for the combination of Trastuzumab and Buparlisib [90]. On the contrary, Buparlisib with Lapatinib showed antitumor activity with a high disease control rate (79%) [91]. Actually, research efforts are focusing on alfa-specific PI3K inhibitors (Taselisib or Alpelisib) in association with anti-HER2 agents (Table 2) [92]. In particular, a phase III trial (EPIK-B2) with alpelisib compared to placebo in combination with trastuzumab and pertuzumab as maintenance treatment after 1st line therapy with trastuzumab, pertuzumab and taxane, in PIK3CA-mutated tumors, is recruiting patients [93]. In a phase I trial, Alpelisib tolerability was tested in combination with trastuzumab emtansine (TDM-1) in trastuzumab-resistant patients [92]. In this study, the combination of alpelisib 250 mg daily and T-DM1 appeared to be safe, with an ORR of 43% in the overall population. Of note, enrolled patients were not selected based on PIK3CA status [92].

## 4. Conclusions

The PI3K/AKT/mTOR pathway is frequently mutated in BC, mainly in HR+ tumors. At present, several studies have demonstrated that mutations on the PI3K/AKT/mTOR pathway promote treatment resistance. The increasing knowledge of the PI3K/AKT/mTOR molecular pathway provides a new perspective for the management of BC. In particular, combined therapy regimens that inhibit parallel pathway activation (i.e., PI3K inhibitors and HER2-targeted agents) seem to be a valid therapeutic approach. In order to reach the full potential efficacy and avoid overlapping toxicity, the safety profile of these targeted combinations should be carefully taken into account. Robust clinical studies regarding class-related side effects and testing different therapeutic doses, such as intermittent dosing schedules, may be useful in reducing side effects and improving patients’ treatment adherence. 

Currently, in Europe, only two PI3K/mTOR inhibitors (Everolimus and Alpelisib) are available for the treatment of HR+/HER2− MBC patients. However, the use of Alpelisib is limited to PI3K-mutated patients who progressed after endocrine monotherapy alone, restricting its use in daily practice. Even if ESMO guidelines do not recommend genomic profiling for the treatment choice of MBC patients, oncologists have to be aware that an actionable mutation could be useful in future patients’ treatment strategy. Considering the growing body of evidence from ongoing clinical and preclinical trials, new treatment strategies and target drugs will likely emerge in future years.

## Figures and Tables

**Figure 1 biomedicines-11-00109-f001:**
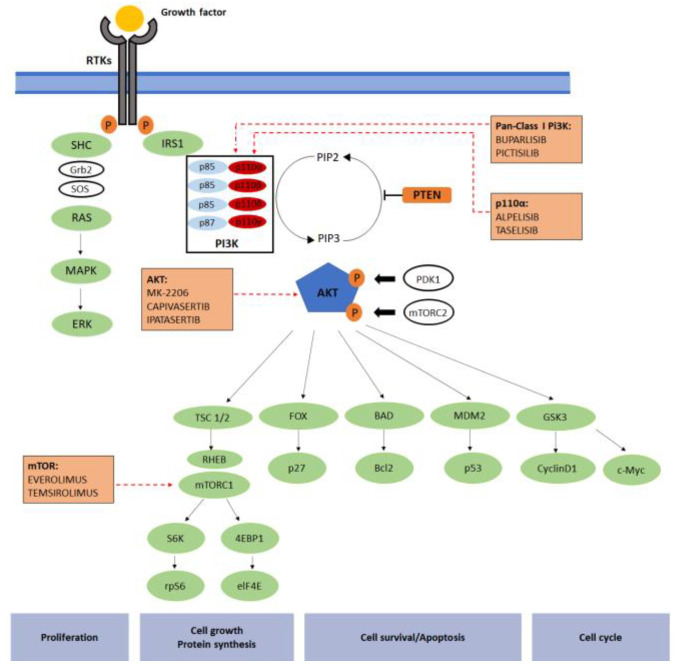
PI3K signaling pathway. The PI3K signaling pathway has a role in mechanisms such as cell growth, survival and metabolism. Following growth factor stimulation and subsequent activation of RTKs, class IA proteins are recruited to the membrane by direct interaction with p85 subunit, with the activated receptors or by interaction with adaptor proteins associated with the receptors. The p110β-containing enzymes might be activated by GPCRs converting PIP2 to PIP3 and providing docking sites for PDK1 and AKT. In particular, PDK1 phosphorylation and AKT activation regulate a downstream signaling event where PTEN is one of the most important final targets. The figure underlines where the multi-, pan- or isoform-specific inhibitors work.

**Figure 2 biomedicines-11-00109-f002:**
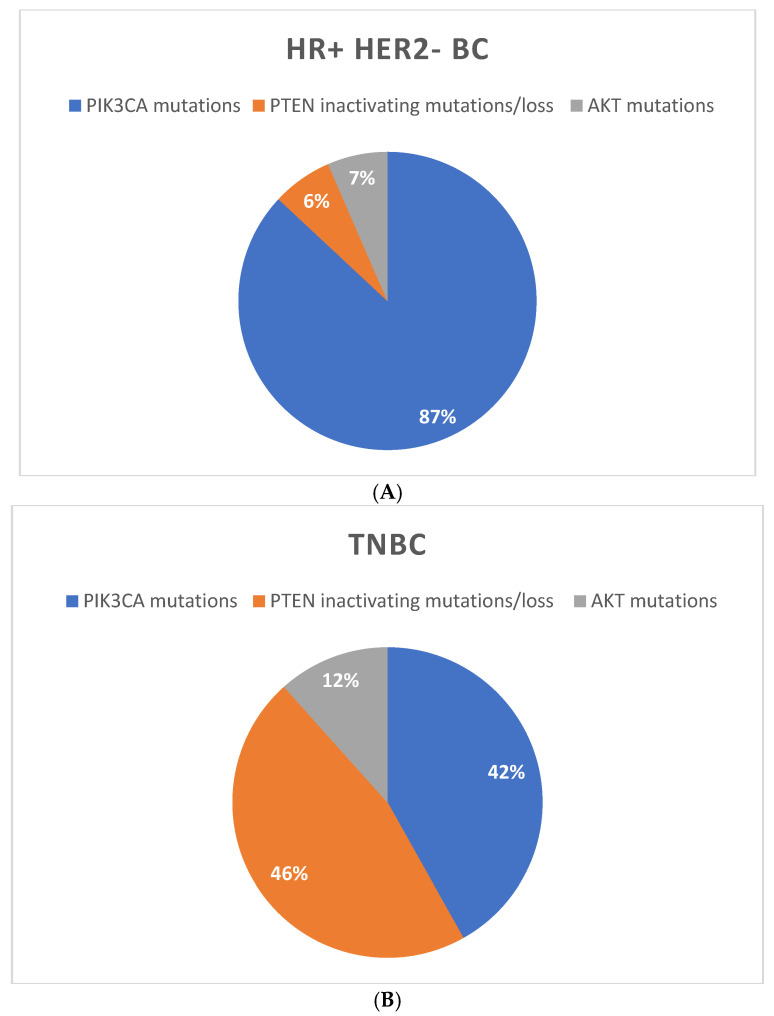
Frequency of PI3K/AKT/mTOR mutations in different breast cancer subtypes: (**A**) HR positive HER2 negative BC, (**B**) TNBC and (**C**) HER2 positive BC, respectively.

**Table 1 biomedicines-11-00109-t001:** PI3K/AKT/mTOR pathway inhibitors.

Class	Target
**Pan I PI3K inhibitors**	
Buparlisib (BKM120)	pan-PI3K
Pictilisib (GDC-0941)	pan-PI3K
Copanlisib (BAY 80-6946)	pan-PI3K
SAR245408 (XL147)	pan-PI3K
PX866	pan-PI3K
**Isoform-specific PI3K inhibitors**	
Taselisib (GDC-0032)	p110α
Alpelisib (BYL719)	p110α
MLN1117	p110α
BAY 1082439	p110α/β
CH5132799	PI3Kα/γ
GSK2636771	p110β
AZD8186	p110β
SAR260301	p110β
Idelalisib (CAL-101)	p110δ
IPI-145	p110δ
AMG319	p110δ
**Dual-specificity PI3K/mTOR inhibitors**	
BEZ235	PI3K/mTOR
GDC-0980	PI3K/mTOR
RF-05212384	PI3K/mTOR
PF-0691502	PI3K/mTOR
GSK-2126458	PI3K/mTOR
SAR245409 (XL765)	PI3K/mTOR
**mTOR inhibitors, rapalogs**	
Sirolimus	mTOR
Nab-rapamycin	mTOR
Temsirolimus	mTOR
Everolimus	mTOR
Radaforolimus	mTOR
**mTOR inhibitors, catalytic**	
OSI-027	mTOR
AZD2014	mTOR
MLN0128	mTOR
PP242	mTOR
ML-223	mTOR
**AKT inhibitors**	
Ipatasertib (GDC-0068)	AKT1/2/3
MK-2206	AKT1/2/4
Capivasertib (AZD5363)	AKT1/2/5
Perifosine (KRX-0401)	AKT1/2/6
GSK2141795	AKT1/2/7
ALM301	AKT1/2
Archexin (RX-0201)	AKT1

**Table 2 biomedicines-11-00109-t002:** Ongoing Trials in BC.

Trial	Phase	Population	Arms
NCT03959891	I	HR+/HER2−	Fulvestrant + Ipatasertib vs. Aromatase Inhibitor + Ipatasertib vs. Fulvestrant + Ipatasertib + Palbociclib
NCT04060862	IB-III	HR+/HER2−	Stage 3: ipatasertib + palbociclib + fulvestrant vs. placebo + palbociclib + fulvestrant
NCT03337724	III	TNBC	ipatasertib + paclitaxel vs. placebo + paclitaxel
NCT03280563	IB-II	HR+/HER2−	Stage 1: Atezolizumab + Ipatasertib + Fulvestrant vs. Atezolizumab + Ipatasertib vs. Atezolizumab + Fulvestrant vs. Atezolizumab + Entinostat vs. Fulvestrant (placebo)Stage 2: Atezolizumab + Bevacizumab + Endocrine Therapy
NCT03800836	I	TNBC	In Cohort 1: ipatasertib + atezolizumab + paclitaxel (nab-paclit) +/− antra
NCT03424005	IB-II	TNBC	Stage 1: Atezolizumab + Nab-Paclitaxel +/− Tocilizumab vs. Nab-Paclitaxel vs. Atezolizumab + Sacituzumab GovitecanStage 2: Capecitabine vs. Atezolizumab + Ipatasertib vs. Atezolizumab + SGN-LIV1A vs. Atezolizumab + Selicrelumab + Bevacizumab vs. tezolizumab + Chemo (Gemcitabine + Carboplatin or Eribulin)
NCT03395899	II	HR+/HER2−	Atezolizumab vs. Atezolizumab + Cobimetinib vs. Atezolizumab + Ipatasertib vs. Atezolizumab + Ipatasertib + Bevacizumab
NCT02390427	I	HER2+	Taselisib + Pertuzumab + Trastuzumab + Paclitaxel vs. Taselisib + Pertuzumab + Trastuzumab vs. Taselisib + Trastuzumab emtansine + Pertuzumab vs. Taselisib + Trastuzumab emtansine
NCT02167854	I	HER2+	Study Evaluating the Safety and Tolerability of LJM716, BYL719 and Trastuzumab in Patients with Metastatic HER2+ Breast Cancer
NCT04208178	III	HER2+	Study of Alpelisib (BYL719) in Combination with Trastuzumab and Pertuzumab as Maintenance Therapy in Patients With HER2-positive Advanced Breast Cancer With a PIK3CA Mutation (EPIK-B2)

Abbreviations: human epidermal growth factor receptor-2 positive (HER2+), hormone receptor positive (HR+), triple-negative breast cancer (TNBC).

**Table 3 biomedicines-11-00109-t003:** Clinical trials of PI3K/AKT/mTOR inhibitors in PI3k-mutated early breast cancer.

Trial	Phase	Population	Arms	pCR Rate %
LORELEI	II	HR+/ HER2−	Letrozole + taselisib vs. letrozole + placebo	pCR:2 %2 taselisib vs. 1% placebo *p* = 0.37
NEO ORB	I	HR+/ HER2−	alpelisib + letrozole vs. letrozole + placebo	pCR: 1%alpelisib vs. 2% placebo *p* = 0.282
FAIRLANE	II	TNBC	paclitaxel + ipatasertib vs. paclitaxel + placebo	pCR: 18% ipatasertib vs. 12% placebo, *p* = NA
I-SPY 2 trial	I	HER2+	MK-2206 + paclitaxel + trastuzumab vs. MK-2206 + paclitaxel vs. paclitaxel vs. paclitaxel + trastuzumab	pCR: 48% MK-2206 arm vs. 29% placebo, *p* = NA
NEOPHOEBE	II	HER2+	buparlisib + trastuzumab vs. placebo + trastuzumab	pCR: 32% buparlisib vs. 40%; *p* = 0.811

Abbreviations: pathological complete response (pCR), Overall response rate (ORR), human epidermal growth factor receptor-2 positive (HER2+), hormone receptor positive (HR+), triple-negative breast cancer (TNBC), not available (NA).

**Table 4 biomedicines-11-00109-t004:** Phase II/III trials of PI3K/AKT/mTOR inhibitors in advanced breast cancer.

Study	Phase	Population	Arms	mPFS, Months
SOLAR-1	III	HR+/ HER2−	Alpelisib + FLV vs. placebo + FLV	*PIK3CA not mut*: mPFS 7.4 alpelisb vs. 5.6 placebo HR 0.85 *PIK3CA mut*: mPFS 11.0 alpelisib vs. 5.7 placebo, *p* < 0.001
BELLE-2	III	HR+/ HER2−	Buparlisib + FLV vs. placebo + FLV	mPFS 6.9 buparlisib vs. 5.0 placebo *p* < 0.001 *PIK3CA-mut*: mPFS 7 buparlisib vs. 3.2 placebo, *p* < 0.001
BELLE-3	III	HR+/ HER2−	Buparlisib + FLV vs. placebo + FLV	mPFS 3.9 buparlisib vs. 1.8 placebo, *p* = 0.00030
BELLE-4	II/III	HR+/ HER2−	Txl + buparlisib vs. Txl + placebo	mPFS 9.2 Buparlisib vs. 8.0 placebo + paclitaxel, HR 1.18
FERGI	II	HR+/ HER2−	Pictilisib + FLV vs. placebo + FLV	mPFS Part 1: 6.6 Pictilisib vs. 5.1 placebo, *p* = 0.096 mPFS Part 2: 5.4 Pictilisib: vs. 10, *p* = 0.84
SANDPIPER	III	HR+/ HER2−	taselisib + FLV vs. placebo + FLV	mPFS 7.4 taselisib vs. 5.4, *p* = 0.0037
TAMRAD	II	HR+/ HER2−	TAM + everolimus vs. TAM alone	mPFS 8.6 everolimus vs. 4.5 placebo, *p* = 0.002
BOLERO-2	III	HR+/ HER2−	EXE + everolimus vs. Exe + placebo	mPFS 6.9 everolimus vs. 2.8 placebo, *p* < 0.001
FAKTION	II	HR+/ HER2−	Capivasertib + FLV vs. FLV + placebo	mPFS 10.3 capivasertib vs. 4.8 placebo, *p* = 0.0018
PEGGY	II	HR+/ HER2−	Txl + pictilisib or Txl + placebo	mPFS 8.2 Pictilisib vs. 7.8 placebo, *p* = 0.83
HORIZON	III	HR+/ HER2−	LET + temsirolimus vs. LET + placebo	mPFS 8.9 temsirolimus vs. 9 placebo, *p* = 0.5
LOTUS	II	TNBC	Txl + ipatasertib vs. Txl + placebo	mPFS 6.2 ipatasertib vs. 4.9, *p* = 0.037 *PTEN-low*: mPFS 6.2 months ipatasertib vs. 3.7, *p* = 0.18
PAKT	II	TNBC	Txl + capivasertib vs. txl + placebo	mPFS 5.9 capivasertib vs. 4.2 placebo, *p* = 0.06 *PIK3CA/AKT1/PTEN-altered*: mPFS 9.3 capivasertib vs. 3.7 placebo, *p* = 0.01

Abbreviations: advanced breast cancer (ABC), pathological complete response (pCR), Overall response rate (ORR), Median progression free survival (mPFS), human epidermal growth factor receptor-2 positive (HER2+), hormone receptor positive (HR+), triple-negative breast cancer (TNBC), Fulvestrant (FLV), Exemestane (EXE), Tamoxifen (TAM), Taxol (Txl).

## Data Availability

This study did not report any new data.

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
