# Peer review of "Targeting PI3K/AKT/mTOR Pathway in Breast Cancer: From Biology to Clinical Challenges"

_biomedicines, 2023, doi:10.3390/biomedicines11010109_

Round 1
Reviewer 1 Report
In this review, the authors present treatment efficacy and clinical outcomes showing that the inhibition of PI3K/AKT/mTOR is effective in BC with PI3K/AKT/mTOR mutations. The authors propose that PI3K/AKT/mTOR inhibitor treatment may be a promising therapeutic strategy in BC. There are additional issues to address to improve the depth of the study and translational potential. Considering the presence of activating PIK3CA mutations in approximately 30% of HER2+ tumors and how PIK3CA mutations plays a major role in with resistance to HER2 inhibitors, there is an urgent need to develop combination regimens blocking HER2 and PI3K. Combination of Trastuzumab and Pertuzumab has shown efficacy in clinical trials and is now approved for the treatment of patients with HER2+ Breast Cancer.
These are some of the studies that can also be considered
1. The ongoing study of Alpelisib in Combination with Trastuzumab and Pertuzumab in patients with PIK3CA mutant HER2+ Advanced Breast Cancer.
2. The authors can also discuss ADC (T-DM1) such as the combination of alpelisib and T-DM1 as it is tolerable and has activity in patients with trastuzumab-resistant HER2+ MBC.
3. To reach the full potential of PI3K inhibitors, some suggestions may be (i) testing therapeutic dosing schedules that may cause less toxicity, and (ii) combining therapies based on robust preclinical studies.
The studies documented in this review are interesting and potentially relevant to disease treatment. Potential combination studies with other targeted agents or conventional treatments would also be of interest.
Reviewer 2 Report
A timely review article by Dr. Omarini and group elaborating on the role of PI3K/AKT/mTOR pathway in breast cancer and also discusses the relevance of its therapeutic impact of it. It is a very well-written article that discusses the translational impact of this PI3K/mTOR pathways in breast cancer. A few things need to be addressed before it is ready for acceptance. They are as follows:
1. It has been shown that AMPK regulates mTOR in a feedback loop manner, and this opens up the possibility of utilizing AMPK activators (e.g., metformin, AICAR) in combination with rapamycin for cancer cells (PMID: 26323019 and PMID: 30107094). This point should be discussed in this review. It will be an important aspect to be discussed briefly.
2. It has been widely discussed the different dosages of rapamycin and how it affects mTORC1 and mTORC2 complexes differentially (PMID: 26916116 and PMID: 24508508). These points must be touched upon by adding a few lines.
3. Another important aspect that must be discussed is that PLD inhibitors might be another option that can be utilized as another alternative to mTOR inhibitors. This has been shown in (PMID: 18927511 and PMID: 31767684). The authors should add a few lines on this point.
4. Lastly, it has been discussed how mTOR plays a role in metabolic checkpoint regulators, and that opens up the opportunity of using metabolic inhibitors (e.g., glutamine metabolic pathways inhibitors) along with mTOR inhibitors (PMID: 26682255 and PMID: 30131808). This novel aspect should be discussed by adding a few lines on this aspect.
Round 2
Reviewer 2 Report
All concerns have been addressed- ready for acceptance.